# Part-level Reconstruction for Self-Supervised Category-level 6D Object Pose Estimation with Coarse-to-Fine Correspondence Optimization

## ABSTRACT

Self-supervised category-level 6D pose estimation stands as a fundamental task in computer vision. However, current self-supervised methods face two major challenges. Firstly, existing networks struggle to reconstruct precise object models due to significant part-level shape variations among specific categories. Secondly, they are impacted by the many-to-one ambiguity in the correspondences between pixels and point clouds. To address these challenges, we propose a novel approach that includes a Part-level Shape Reconstruction (**PSR**) module and a Coarse-to-Fine Correspondence Optimization (**CFCO**) module. In the (**PSR**) module, we introduce a part-level discrete shape memory to capture more fine-grained shape variations of different objects and use it to perform precise reconstruction. In the (**CFCO**) module, we utilize Hungarian matching to generate one-to-one pseudo labels at both region and pixel levels, which provides explicit supervision for the corresponding similarity matrices. We evaluate our method on the REAL275 and WILD6D datasets. Our extensive experiments show that our method outperforms existing methods and achieves new state-of-the-art results.

## CCS CONCEPTS

• **Computing methodologies → Computer vision**.

## KEYWORDS

Deep Learning, Multimodal Data Processing, 3D Reconstruction, Self-Supervised Learning, Visual-Spatial Correspondence

## 1 INTRODUCTION

6D object pose estimation, which involves estimating the 3D translation and 3D rotation of objects, has played a pivotal role in domains like autonomous driving [23], virtual reality [2] and augmented reality [10]. Traditional instance-level pose estimation can only estimate poses for specific instances, resulting in poor generalization. In contrast, category-level pose estimation methods achieve great generalization across different instances within the same category by utilizing categorical priors and training on large-scale datasets.

Permission to make digital or hard copies of all or part of this work for personal or classroom use is granted without fee provided that copies are not made or distributed for profit or commercial advantage and that copies bear this notice and the full citation on the first page. Copyrights for components of this work owned by others than the author(s) must be honored. Abstracting with credit is permitted. To copy otherwise, or republish, to post on servers or to redistribute to lists, requires prior specific permission and/or a fee. Request permissions from permissions@acm.org.

*ACM MM, 2024, Melbourne, Australia*

© 2024 Copyright held by the owner/author(s). Publication rights licensed to ACM.

ACM ISBN 978-x-xxxx-xxxx-x/YY/MM

https://doi.org/10.1145/nnnnnnn.nnnnnnn

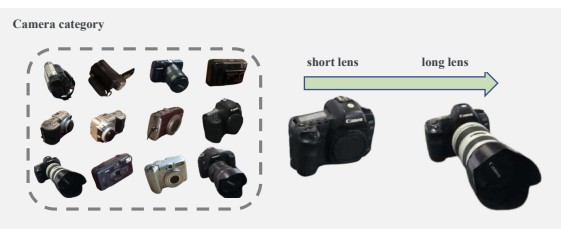

**a) Significant shape variations in local parts**

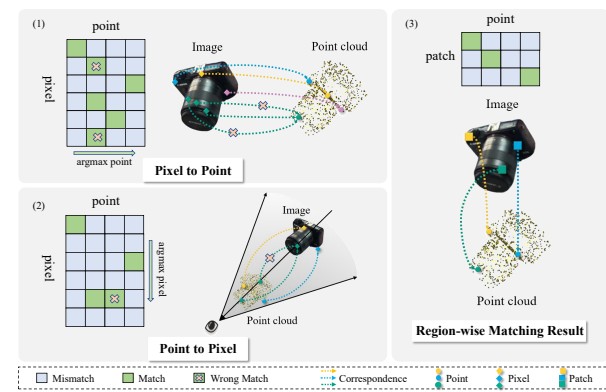

**b) Many-to-one ambiguity**

**Figure 1: a) Objects that belong to the same category often have similar overall structures but exhibit significant shape variations in local parts. b) The many-to-one ambiguity occurs when there are more image pixels than points in the point cloud model. This ambiguity can also be caused by projection issues, where the model's rear cannot be displayed in the image. To eliminate these ambiguities, the proposed region-level relationships generated through Hungarian matching can achieve a one-to-one correspondence.**

However, annotating 6D labels for such real-world datasets is time-consuming. Therefore, self-supervised methods have been proposed to address this issue.

[34] explores self-supervised category-level pose estimation for the first time. They first reconstruct instance models based on categorical shape priors. Then correspondences between pixels and reconstructed models are established by calculating the pixel-point similarity matrix. Additionally, they propose a cycle-consistency loss to impose implicit supervision on the similarity matrix. To achieve accurate pose estimation without 6D labels, we summarize that two critical problems should be carefully considered as shown in 1: 1) In the task of category-level pose estimation, instances within the same category share similar overall structures, but often

exhibit significant shape variations in local parts. Previous methods struggle to perform precise reconstructions in these regions. We argue that is because they rely on a fixed categorical shape prior which is insufficient to represent such local variations, particularly in unsupervised scenarios. 2) Since the number of pixels in images is significantly greater than that of points in 3D models, it is inevitable that multiple pixels will be mapped to same points, causing many-to-one ambiguity in pixel-to-point correspondences. On the other hand, 2D images only show the visible part of model points. Thus, the invisible model points may be mapped to those pixels corresponding to the visible points, leading to many-to-one ambiguity. The aforementioned many-to-one relationship between 2D pixels and 3D points can degrade the network's performance. Since the correspondences between 2D pixels and 3D points are established using the pixel-point similarity matrix, optimizing this matrix is of significant importance.

Based on aforementioned discussions, we propose a novel self-supervised category-level 6D object pose estimation method, which comprises two key modules: a **P**art-level **S**hape **R**econstruction (**PSR**) module using adaptive shape memory and a **C**oarse-to-**F**ine **C**orrespondence **O**ptimization module (**CFCO**). In the **PSR** module, we use a discrete shape memory to capture more detailed shape variations of different objects at the part-level. We start by extracting patch features from images using an image encoder and then retrieve their nearest shape codes from the discrete shape memory based on feature similarity. After that, we use a shape decoder to generate 3D object models based on the selected shape codes. To ensure the network is well-trained, we pre-train our shape reconstruction module on the ShapeNet dataset with a large number of image-shape pairs. In the **CFCO** module, we employ the Hungarian matching algorithm to generate one-to-one pseudo labels, facilitating a coarse-to-fine supervision approach at the region and pixel levels, providing explicit supervision for the corresponding similarity matrices. When it comes to point clouds, they cover more extensive areas with fewer points, leading to a difference in density and a significant disparity in spatial granularity. To reconcile this granularity inconsistency between images and point clouds, we initially compute a region-level similarity matrix using intermediate image features and point cloud features that have similar granularity. We use the Hungarian matching algorithm to process this similarity matrix, filtering out invisible points and establishing more precise one-to-one correspondences. These one-to-one correspondences serve as coarse pseudo labels to supervise the regional similarity matrix. To achieve more refined and accurate correspondences between pixels and points, we generate fine-grained supervision by identifying the most similar pixel within each region. This conversion of the initial coarse region-level pseudo-labels to more detailed ones allows us to produce more refined and accurate correspondences between pixels and points, thereby significantly enhancing the accuracy of our pose estimation. In summary, the primary contributions of our work are listed as follows:

- We introduce a novel method for self-supervised category-level 6D object pose estimation, which comprises two crucial modules: the **CFCO** module and the **PSR** module with Adaptive Shape Memory. They collaborate to better perform in the self-supervised setting with large shape variations.

- In the Coarse-to-Fine Correspondence Optimization module, a one-to-one region correspondence is provided as explicit supervision for both coarse-grained and fine-grained similarity matrices through global optimization. In the Part-level Shape Reconstruction module, discrete codebooks are utilized to capture part-level semantic features of objects, adapting to reconstruct point clouds for each image.

- Our proposed method achieves state-of-the-art results on the NOCS dataset. Furthermore, we conduct comprehensive ablation experiments to verify the effectiveness of our designs.

## 2 RELATED WORK

**Fully-supervised Pose Estimation.** Fully-supervised 6D pose estimation can be categorized into instance-level [13, 17, 20, 28, 29] and category-level methods [3, 24, 30]. Given specific models for each instance, methods such as BB8 [20] and PVNet [17] predefine a set of keypoints. These networks retrieve pose by establishing correspondences among these keypoints. Additionally, there are instance-level methods that directly regress pose [28] or establish dense prediction [29]. To achieve generalization across instances within a category, category-level methods incorporate category priors. [30] introduced Normalized Object Coordinate Space (NOCS), which is later aligned with diverse objects to retrieve pose. Utilizing the estimated NOCS, the 6D pose is then determined through the application of the Umeyama algorithm [25]. Subsequent research endeavors [3, 24] have emerged, focusing on refining and enhancing the accuracy of NOCS representations.

**Semi/Self-supervised for Pose Estimation.** Due to the substantial labor and time costs for annotating 6D labels in the real-world, various semi/self-supervised methods have been proposed to enhance generalization. Some works are based on semi-supervised learning [27, 33], where networks are trained initially on labeled virtual datasets and unsupervised real-world datasets. These approaches all involve pretraining on virtual datasets to acquire favorable initialization. Besides, some of the semi-supervised focus on migitating sim-to-real domain gap [5, 11, 12, 14]. To further address the issue of labeled data requirement, self-supervised methods are employed [34], whose networks are indirectly supervised through 2D/3D labels. Our proposed method addresses the challenge of indirect supervision by embedding a pseudo-label generation module that progresses from coarse to fine.

**VQVAE for generation.** In generative tasks, VQVAE [26] has been widely employed. VQVAE [26] learns the discrete representation of image patches and models their distribution. VQVAE2 [21] extends the original VQVAE model with a hierarchical discrete representation. In addition to 2D generative tasks, some studies have utilized VQVAE methods in the 3D domain for 3D reconstruction tasks. Canonical Mapping [6] proposes a model using VQVAE to generate point clouds. In concurrent research, AutoSDF [16] generalizes VQVAE to 3D voxels. To capture more fine-grained shape variations of different objects, we introduce a part-level discrete shape memory based on VQVAE.

Figure 2: Framework Overview. Our method is comprised of two primary modules: the Part-level Shape Reconstruction Module and the Coarse-to-Fine Correspondence Optimization Module. The Part-level Shape Reconstruction Module adaptively learns part features through a discrete shape memory $e$, effectively reconstructing the object model $P$. In the Coarse-to-Fine Correspondence Optimization Module, a regional similarity matrix $S_c$ is computed between the image features $F_2$ and the point features $F_{geo}$ and is then refined using coarse pseudo-labels $T_c^*$, which are obtained via Hungarian Matching. Subsequently, the fine pseudo-labels $T_f^*$, derived from $T_c^*$ through index-based upsampling, supervise the pixel-point similarity matrix $S_f$, enhancing the accuracy of correspondence at a finer granularity.

## 3 METHOD

Our method aims to infer the rotation $R \in \mathbb{R}^{3 \times 3}$ and translation $T \in \mathbb{R}^3$ of objects solely from RGB images, without relying on annotated object poses. Figure 2 illustrates the workflow of our method, which is divided into two main stages. In the first stage, we pre-train our **PSR** module on the ShapeNet dataset. Using synthetic RGB images $x$, we reconstruct the corresponding 3D point cloud model $P$ of the object depicted in the image. This stage serves to provide a robust initialization for our part-level code memory $e \in \mathbb{R}^{K \times C}$, which contains $K$ shape codes. Leveraging ShapeNet's extensive range of 3D models enhances the robustness and accuracy of our method (see Section 3.1). In the second stage, we engage in self-supervised training using unlabeled data. Initially, the object model $P \in \mathbb{R}^{N \times 3}$ is reconstructed through **PSR** module using the pre-trained reconstruction module with the learned shape memory. We then employ a PointNet[19] to extract geometric features $F_{geo} \in \mathbb{R}^{N \times C}$ from the reconstructed $P$, and a U-Net[22] to extract multi-level features $\{F_l\}_{l=0}^3$ from the images. Subsequently, we compute the similarity between $F_{geo}$ and the multi-level image features $F_2$ and $F_3$, which are derived from the penultimate and last layers of the U-Net, producing similarities at both regional and pixel levels. Furthermore, The **CFCO** module processes the computed similarities to generate pseudo-labels, which in turn are used to explicitly supervise the similarities at various granularities. This step is crucial for enhancing the precision of the correspondence establishment, ensuring that our method can accurately map the

3D object features to the corresponding features in the RGB images. (see Section 3.2). we will detail each part in the following sections.

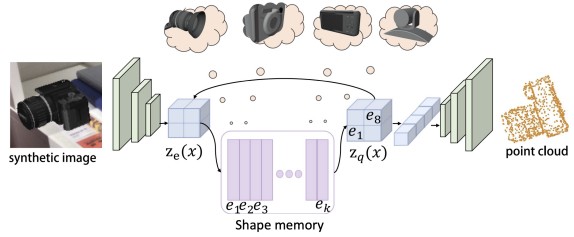

Figure 3: An illustration of our part-level shape reconstruction module. Our proposed reconstruction model captures part-level shape variations for a better reconstruction

### 3.1 Part-level Shape Reconstruction with Adaptive Shape Memory

To provide a good initialization for the reconstruction module, we first conduct pretraining on a large-scale dataset ShapeNet [30]. Since ShapeNet consists of abundant point cloud data, we leverage them to render a substantial number of images to pre-train our reconstruction module. Given a rendered image $x$, let $F$ denote the feature map extracted from the image encoder. The feature map is then patchified into $N$ patches, which are then subject to average pooling to obtain region-aware descriptors denoted as

$z_e(x) \in \mathbb{R}^{N \times C}$. This method of patchifying and pooling is instrumental in enhancing the model's sensitivity to local variations in the image. By breaking down the feature map into smaller, manageable regions, our model can focus on fine details within each segment, thereby capturing subtle differences that may be lost in a more global analysis.

Concurrently, we establish a latent code memory $e \in \mathbb{R}^{K \times C}$, where $K$ represents the number of discrete shape codes. The decoding process involves matching each descriptor to the closest shape code in the memory, as defined by equation 1:

$$z_q(x) = e_k, \quad k = \arg\min_j \|z_e(x) - e_j\|_2. \tag{1}$$

These randomly initialized shape codes are refined during training through a Nearest Neighbor matching mechanism to effectively capture diverse local shape features. The use of a discrete latent space, as opposed to a continuous one, results in more stable training dynamics and higher quality of the generated models. This discrete representation allows for the precise encoding of complex geometrical information and avoids the over-smoothing often associated with continuous autoencoders. Furthermore, the discrete nature ensures the consistent and robust selection of shape codes, enhancing the model's ability to learn distinct and intricate patterns.

After retrieving shape codes $z_q(x)$ for $N$ patches, we concatenate them into a global feature vector $g \in \mathbb{R}^{1 \times NC}$, which is then fed into a three-layer MLP to reconstruct the 3D shape. We employ the Chamfer Distance as the reconstruction loss, effectively measuring the shape discrepancy between the reconstructed point cloud $P$ and target point cloud $\hat{P}$. In detail,

$$D_{cd}(P, \hat{P}) = \frac{1}{|P|} \sum_{x \in P} \min_{y \in \hat{P}} \|x - y\|^2 + \frac{1}{|\hat{P}|} \sum_{y \in \hat{P}} \min_{x \in P} \|x - y\|^2. \tag{2}$$

Similar to the approach taken in [26], we have addressed the issue of undefined real gradient for equation 1 that is common in VQ-VAE architectures. To do this, we have updated the weights of the encoder and the code memory using a straight-through estimator [1] and moving averages, respectively. As a result, the final loss function is composed of three components, which are as follows:

$$L_{recon} = D_{cd}(P, \hat{P}) + \|sg[z_e(x)] - e\|_2^2 + \beta \|z_e(x) - sg[e]\|_2^2, \tag{3}$$

where $\beta$ is a hyperparameter, and $sg$ denotes the stop-gradient operation.

After obtaining the pre-trained model, we introduce the shape memory into the self-supervised network and fine-tune both the encoder and the shape memory to better adapt to real images. During this process, real images are transformed into latent representations through the encoder and then compared with the vectors in the pre-trained shape memory to identify the most matching shape code. The selected shape codes are subsequently fed into the pre-trained decoder, ultimately generating the reconstructed point cloud $P$. In this way, the pre-trained shape memory serves as a crucial component to capture the local shape variations of different instances, enhancing the quality of the reconstructed shapes.

## 3.2 Coarse-to-Fine Correspondence Optimization

**Coarse correspondence estimation.** To meditate the significant disparity in the spatial granularity between pixels and points, we first establish region-level correspondences by matching intermediate patch features of the image with point features. Specifically, given a reconstructed object model $P \in \mathbb{R}^{N \times 3}$, we employ a feature extractor to obtain point-wise features $V \in \mathbb{R}^{N \times C}$ where $C$ is the feature dimension. Considering the sparsity of the point cloud and the aggregation of surrounding point information during feature extraction, we treat each point as a descriptor of regional features. Simultaneously, we encode the input image into multi-level features $\{F_l\}_{l=0}^3$ using a U-Net architecture, where the encoder part is the ResNet18 pretrained as described in section 3.1. Specifically, we utilize the features $F2$, corresponding to the penultimate layer, to represent image regions. Then, we compute the coarse similarity matrix $S_c \in \mathbb{R}^{h_2 w_2 \times N}$ between these image regions and point cloud regions using cosine similarity, where $h_2, w_2$ denote the height and width of the image features $F_2$. In detail,

$$S_c = \frac{F_2 \cdot V}{\|F_2\|\|V\|}. \tag{4}$$

**Coarse mapping planner.** To enhance the learning of the region-level similarity matrix, we propose to generate one-to-one coarse pseudo labels $T_c^* \in \mathbb{R}^{h_2 w_2 \times N}$ for ensuring a unique correspondence between each image region and point cloud region. The pseudo label assignment process is the same as the Optimal Transport (OT), and the goal of the OT problem is to find a transportation plan at a global minimal transportation cost, which can be solved by the Hungarian Matching algorithm. Specifically, given the point region features $V$, we aim to map the image region features $F_2$ to them. We denote this regional mapping as $T_c^*$ with a similarity cost matrix $(1 - S_c) \in \mathbb{R}^{h_2 w_2 \times N}$. The optimal transportation plan $T_c^*$ is obtained by minimizing the similarity cost:

$$T_c^* = \arg\min_{T \in \mathcal{T}} \sum_{i=1}^N \sum_{j=1}^N T_{ij} \cdot C_{ij}, \tag{5}$$

Here, due to the image features being fewer than the point cloud features, we introduce $N - hw$ virtual rows to the similarity matrix to equalize the number of rows and columns for the standard Hungarian matching, discarding these virtual rows afterward. $\mathcal{T}$ is the search space limited as:

$$\mathcal{T} = \left\{ \mathcal{T} \in \mathbb{R}^{N \times N} \mid T\mathbf{1} = \frac{1}{N} \cdot \mathbf{1}, \quad T^T\mathbf{1} = \frac{1}{N} \cdot \mathbf{1} \right\}, \tag{6}$$

where $\mathbf{1}$ denotes the vector of all ones. To online refine the similarity matrix, the pseudo label loss is imposed to make the similarity, which is computed as:

$$L_c = CE(T_c^*, S_c), \tag{7}$$

In essence, after aligning the densities of images and point clouds using image region representations, the number of regions in the image is less than that in the point cloud. The Hungarian matching process, by discarding virtual rows of the image, helps to ignore the invisible parts of the point cloud, effectively solving the inherent

many-to-one challenge of mapping point clouds to image pixels and significantly enhancing the precision of 6D pose estimation.

**Fine correspondence estimation.** Employing the aforementioned approach, we enhance the regional-level matching between images and point clouds, initially establishing a coarse region-level correspondence. To refine the pose estimation, we aim to derive a more precise fine-matching relationship, which involves specific pixel positions to point cloud coordinates. Initially, we calculate a pixel-level similarity matrix $S_f \in \mathbb{R}^{h_3 w_3 \times N}$ by evaluating the cosine similarity of each pixel point with every point in the point cloud, where $h_3 w_3$ refer to the height and width from the $F_3$.

**Fine mapping planner.** We generate fine pseudo labels $T_f^*$ from coarse pseudo labels $T_c^*$ through index-based upsampling. We then utilize these fine pseudo labels $T_f^*$ to supervise the pixel-level similarity matrix $S_f$. To articulate this process more clearly, we equate it to initially down-sampling the pixel-level similarity matrix $S_f$ using index-based down-sampling to produce $S_{fc}$, followed by supervising $S_{fc}$ with coarse pseudo labels $T_c^*$. Next, we detail the index-based down-sampling process. Initially, we reshape the similarity matrix $S_f$ into dimensions $h_3 \times w_3 \times n$. Then, we apply max pooling across the spatial dimensions $h_3 \times w_3$ for each of the $n$ feature channels, resulting in a reduced matrix of size $h_2 \times w_2 \times n$. This matrix is further reshaped to $h_2 w_2 \times n$, denoted as $S_{fc}$. We supervise $S_{fc}$ using the coarse pseudo labels $T_c$. In summary, we supervise the similarity of the points with the highest similarity in each region.

$$L_f = \text{CE}(T_f^*, S_{fc}), \tag{8}$$

Thus, the overall loss for coarse-to-fine correspondence optimization is

$$L_{c2f} = L_c + L_f. \tag{9}$$

### 3.3 Inference and Training

We employ a two-stage training strategy. During the first stage, we pre-train a shape reconstruction module on the ShapeNet. In the self-supervised learning phase, we fine-tune this module in an end-to-end manner. Additionally, we incorporate the $L_{cycle}$ to optimize the similarity matrix similar to [34]. The final loss function is as follows:

$$L = L_{c2f} + L_{cycle} \tag{10}$$

In the inference phase, we extract part-level features to look up the shape memory for adaptive model reconstruction and obtain a reliable similarity matrix through the coarse-to-fine constraints employed during training. We then select reliable point pairs using the cycle matching distance method and finally solve for the pose using the UMEYAMA algorithm[25].

## 4 EXPERIMENTS AND RESULTS

### 4.1 Experiments Settings

**Datasets.** The main body of our experiments is conducted on the Wild6D dataset [33], comprising a diverse collection of 5,166 videos featuring 1722 different objects in 5 categories (i.e., bottle, bowl, camera, laptop, and mug). The training set of Wild6D provides RGB-D inputs and object foreground segmentation masks generated using Mask R-CNN [8]. The test set includes 6D pose labels

annotated by humans. In our evaluation of this dataset, we compare our method with the current state-of-the-art semi-supervised method RePoNet [33] and self-supervised approach [34], as well as several pre-trained supervised methods adapted to this dataset.

We also train and conduct evaluations on the commonly used category-level benchmark: NOCS-REAL275. NOCS-REAL275 is a real-world dataset consisting of 6 categories (i.e., bottle, bowl, camera, can, laptop, and mug) across 13 different scenes. A total of 4250 images from each scene are utilized for training, and the remaining 2750 images are reserved for validation. It is worth noting that we pre-train our part-level shape reconstruction module on ShapeNet.

**Evaluation metrics.** As for the Wild6D and NOCS-REAL275 dataset, we report the mean Average Precision (mAP) of 5°2cm, 5°5cm, 10°2cm, 10°5cm metrics. $n°m$ cm denotes the percentage of prediction with rotation prediction error within $n$ degrees and translation prediction error within $m$ centimeters. We also report mAP of 3D Intersection over Union (IoU) at the threshold of 25% and 50%.

**Implementation details.** Our PSR module is initially pre-trained on ShapeNet to establish a good initialization. The entire network is further trained on Wild6D in an end-to-end manner later. We first resize the images to $256 \times 256$ pixels and set the number of point clouds for the reconstruction model to 2048. PointNet is employed as the feature extractor for the point cloud modality, while ResNet18 serves as the feature extractor for the imaging modality. We employ the Open3D library to generate meshes directly from 3D point clouds, which aids in the rendering process. The network is trained on eight NVIDIA RTX3090 GPUs, with a batch size of 16, trained over 100 epochs.is

### 4.2 Comparison with State-of-the-Art Methods

We categorize current pose estimation methods into three groups: 1) fully-supervised methods: with both labeled data in synthetic and real-world datasets; 2) semi-supervised methods: with labeled data only on synthetic dataset; 3) self-supervised methods: without any labeled data both on synthetic and real-world datasets.

**Evaluation on Wild6D.** The quantitative results on the Wild6D dataset are illustrated in the Table 1. The results demonstrate that our method outperforms all previous state-of-the-art (SOTA) methods, achieving an improvement of 2.7% and 5% under 5°2cm and 10°5cm respectively. The enhanced performance of our method can be attributed to a more comprehensive and explicit understanding of object-specific local semantic information. This is achieved through explicit supervision in part-level shape reconstruction and correspondence establishment. For instance, to infer the shape of the camera lens, the network can choose a semantic code controlling this attribute to make more precise predictions. A visualization of pose and the reconstructed mesh is shown in Figure 4. Furthermore, we employ a pseudo-label supervision for correspondence in a coarse-to-fine manner while the deformation networks of the previous methods lack sufficient attention to fine-grained object parts, leading to the superior performance of our method. We visualize points on the image and the 3D point cloud model in each point's features, finding that our method concentrates and accurately captures similarities, as seen in Figures 5(a), 5(b). This visual

| Method Name | Data | IoU25 | IoU50 | 5°2cm | 5°5cm | 10°2cm | 10°5cm |
|---|---|---|---|---|---|---|---|
| CASS [3] | C+R | 19.8 | 1.0 | - | - | - | - |
| Shape Prior [24] | C+R | 55.5 | 32.5 | 2.6 | 3.5 | 9.7 | 13.9 |
| DualPoseNet [15] | C+R | 90.0 | 70.0 | 17.8 | 22.8 | 26.3 | 36.5 |
| GPV-Pose [7] | C+R | 91.3 | 67.8 | 14.1 | 21.5 | 23.8 | 41.1 |
| RePoNet [33] | C+W* | 84.7 | **70.3** | 29.5 | 34.4 | 35.0 | 42.5 |
| Self-GC [34] | W* | 92.3 | 68.2 | 32.7 | 35.3 | 38.3 | 45.3 |
| Ours | W* | **94.1** | 69.8 | **35.4** | **38.2** | **40.5** | **50.6** |

**Table 1: Comparison with SOTA methods on Wild6D. In "Data" column, C=CAMERA25, R=REAL275, S=synthetic objects, W=Wild6D, "*"=w/o pose annotation.**

**Figure 4: Qualitative comparisons on the Wild6D dataset: our proposed method versus the state-of-the-art by [34]. The left column displays the estimated poses with bounding boxes on the input 2D images, and the right column shows the 3D instance models reconstructed from those images.**

evidence further underscores the effectiveness of our coarse-to-fine correspondence optimization strategy.

**Evaluation on NOCS-REAL275.** The quantitative results on the NOCS-REAL275 dataset are presented in the Table 2, showing that our method outperforms all previous SOTA methods and surpasses the [34] by 3.2% and 5.2% under 5°5cm and 10°5cm. On the REAL275 dataset, our method's performance is slightly improved compared to the methods under the same setting. This is primarily attributed to the fact that we train our model exclusively on the REAL275 dataset. Since the REAL275 dataset is relatively small, it is challenging to provide abundant training information, leading to a degradation in our performance. A visualization of pose and the reconstructed mesh is shown in Figure 6.

### 4.3 Ablation Studies

**Effects of part-level shape reconstruction module**. We analyze the effectiveness of each sub-pipe of the module by removing/replacing modules from the original network structure (denoted as a full model). We develop four variations for comparison as shown in Table 3. 1) In the *w/o 3D reconstruction* configuration, we eliminate the image-to-3D reconstruction module and instead directly establish correspondences using the clustered center 3D models for each category from the ShapeNet dataset. The use of generic shape priors leads to a significant performance decline because they fail to capture the local shape variations among different instances within the same category. This inaccuracy affects both the computation of similarity and the loss calculation during rendering, compromising the overall efficacy of the model. 2) In the *static shape memory* setting, the shape memory is pre-trained

| Supervision | Method | Data | IoU25 | IoU50 | 5°5cm | 10°5cm |
|---|---|---|---|---|---|---|
| fully-supervised | NOCS [30] | C+R | 84.8 | 78.0 | 10.0 | 25.2 |
| | Shape-Prior [24] | C+R | - | 77.3 | 21.4 | 54.1 |
| | FS-Net [4] | C+R | **95.1** | **92.2** | 28.2 | 60.8 |
| | DualPoseNet [15] | C+R | - | 79.8 | **35.9** | **68.8** |
| Semi-supervised | DISR [18] | C+R* | 83.2 | 73.0 | 19.6 | 54.5 |
| | RePoNet [33] | C+R* | **85.8** | 76.9 | 31.3 | 56.8 |
| | UDA-COPE [11] | C+R* | 84.0 | **82.6** | **34.8** | **66.0** |
| | NAS [5] | S | 15.5 | 1.3 | 0.9 | 2.4 |
| | CPPF [32] | S | 78.2 | 26.4 | 16.9 | 44.9 |
| Self-supervised | SCPE [9] | R*+Y* | 83.5 | **58.7** | 5.6 | 17.4 |
| | REAL[34] | R* | 76.3 | 41.7 | 11.6 | 28.3 |
| | Wild6D[34] | W* | 89.3 | 49.5 | 13.7 | 33.7 |
| | Ours-REAL | R* | 80.1 | 44.5 | 14.7 | 30.4 |
| | Ours-Wild6D | W* | **92.5** | 51.4 | **16.9** | **38.9** |

Table 2: Comparison with SOTA methods on REAL275. In "Data" column, C=CAMERA25, R=REAL275, S=synthetic objects, Y=YCB [31], W=Wild6D, "*"=not using pose annotation.

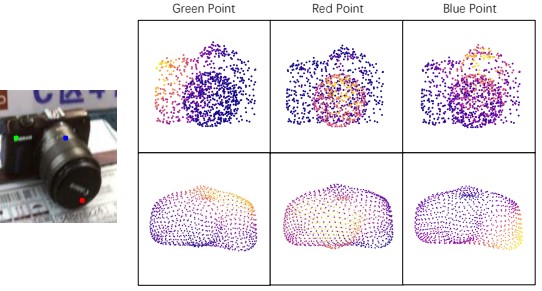

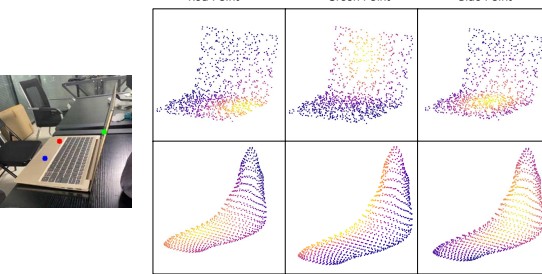

(a) The similarity between individual pixel points and the point cloud (camera)    (b) The similarity between individual pixel points and the point cloud (laptop)

Figure 5: We visualize the similarity between individual pixels in the input image and points in the point cloud. In the visualizations, yellow indicates high similarity while blue represents low similarity. The first row presents results from our method, and the second row shows results from the method described in [34].

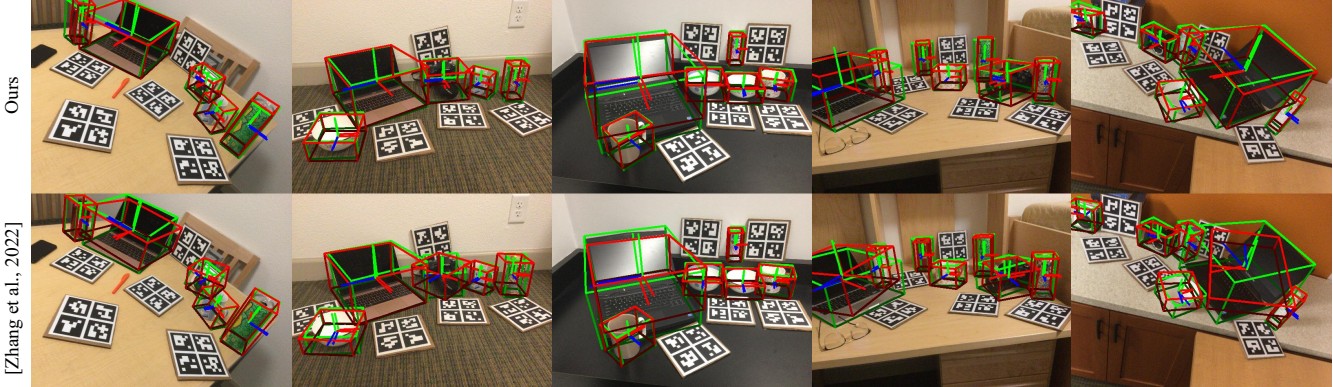

Figure 6: Qualitative comparisons between the state-of-the-art method of [34] and our proposed one on NOCS dataset.

| Method | IoU25 | IoU50 | 5°2cm | 5°5cm | 10°2cm | 10°5cm |
|---|---|---|---|---|---|---|
| w/o 3D reconstruction | 73.2 | 32.4 | 20.7 | 23.6 | 29.4 | 38.1 |
| w/ deformation network | 89.3 | 57.4 | 26.7 | 29.5 | 35.9 | 48.6 |
| w/ static shape memory | 76.5 | 47.6 | 23.4 | 24.0 | 32.1 | 41.9 |
| w/ non-pretrained shape memory | 65.9 | 43.3 | 22.5 | 26.4 | 25.2 | 31.2 |
| w/o correspondences optimization | 90.2 | 65.3 | 31.2 | 35.0 | 36.2 | 47.5 |
| w/o region-level correspondences optimization | 92.3 | 67.6 | 33.6 | 37.5 | 38.5 | 49.2 |
| w/o pixel-level correspondences optimization | 91.2 | 66.3 | 32.3 | 36.3 | 37.6 | 48.9 |
| full model | **94.1** | **69.8** | **35.4** | **38.2** | **40.5** | **50.6** |

**Table 3: Ablation studies on model designs and other settings.**

exclusively on synthetic datasets and then kept frozen during subsequent training on the Wild6D dataset. This approach leads to suboptimal performance due to the simulation-to-reality domain gap, which the static memory cannot bridge effectively.3) In the *w non-prestrained shape memory* setting, shape reconstruction is randomly initiated on the Wild6D dataset without any pre-training, relying solely on self-supervised learning. The absence of direct label supervision, combined with significant shape reconstruction disparities, prevents the rendering module from guiding the optimization correctly, leading to an inability to learn effectively and resulting in suboptimal performance. 4) We also substitute PSR with a deformation network similar to [34]. The results reveal that our module excels in capturing fine-grained details, as traditional deformation networks pay less attention to part-level deformations and rely on a shared category prior to deformation.

**Effects of coarse-to-fine correspondence optimization**. We evaluate the distinct impacts of the CFCO module's two levels of supervision—coarse-grained and fine-grained—by establishing three variations through sequential ablation of these components: 1)*w/o correspondences optimization*: In this configuration, the entire CFCO module is removed for assessment. The absence of CFCO eliminates explicit supervision for all correspondence types, leading to a significant decline in performance. This change distinctly highlights the critical role of the CFCO in providing structured, explicit guidance necessary for accurate correspondence mapping across the entire network. 2)*w/o region-level correspondences optimization*: In this setting, we remove only the coarse-grained, region-level supervision. The result is a marginal improvement, indicating that while region-level correspondences contribute to establishing foundational matches, they alone do not suffice to drive optimal performance across the network. 3)*w/o pixel-level correspondences optimization*: By ablation of the fine-grained, pixel-level supervision, the network experiences a slight degradation in performance. This underlines the importance of pixel-level correspondences in fine-tuning pose estimation accuracy, particularly in enhancing the details and precision provided by region-level correspondences.

## 5 CONCLUSIONS

In this work, we propose a novel network for self-supervised category-level 6D pose estimation. We design a coarse-to-fine correspondence Optimization module and a part-level shape reconstruction module using adaptive shape memory. The first module generates

one-to-one pseudo-labels through the Hungarian matching algorithm to supervise dense correspondence predictions in a coarse-to-fine manner. The second module adaptively learns a part-level shape memory to perform precise reconstructions in local regions of object model. Extensive experiments have demonstrated that our proposed method outperforms the state-of-the-art methods on the REAL275 and Wild6D datasets.

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
