# OpenReview forum: "Part-level Reconstruction for Self-Supervised Category-level 6D Object Pose Estimation with Coarse-to-Fine Correspondence Optimization"
_acmmm.org/ACMMM/2024/Conference — MM2024 Poster_

### Official Review · Reviewer_fj3v · 2024-05-24

**Rating:** 3
**Confidence:** 2

**Summary:**

This paper proposes a self-supervised approach for category-level 6D object pose estimation using two novel modules: Part-level Shape Reconstruction (PSR) and Coarse-to-Fine Correspondence Optimization (CFCO). These modules aim to address challenges in current self-supervised methods, such as significant shape variations and many-to-one ambiguity in correspondences. The method outperforms state-of-the-art techniques on REAL275 and WILD6D datasets.

**Strengths:**

The introduction of part-level discrete shape memory within the PSR module for capturing detailed shape variations is a significant advancement over previous methods that use fixed shape priors. And the method achieves state-of-the-art performance on standard datasets, confirming its effectiveness through extensive ablation studies and comparisons with existing methods.

**Limitations:**

I believe it is best to provide proper citations when mentioning datasets in the paper. Additionally, it is crucial to annotate each metric with directional arrows to indicate whether higher or lower values are preferable. The accuracy and fluency of the writing can also be improved. Furthermore, the equations should be carefully checked (for example, the right side of equation (6) should not be \mathcal{T}).
Lastly, providing access to the source code would greatly enhance the paper's credibility.

**Suitability:**

3

---

### Official Review · Reviewer_vJ9G · 2024-05-24

**Rating:** 4
**Confidence:** 4

**Summary:**

This paper proposes a self-supervised method by a Part-level Shape Reconstruction (PSR) module and a Coarse-to-Fine Correspondence Optimization (CFCO) module. Existing networks struggle to reconstruct precise object models due to significant part-level shape variations among specific categories, and they are impacted by the many-to-one ambiguity in the correspondences between pixels and point clouds. The proposed two modules can solve the above two problems.

**Strengths:**

1. The paper is good writing, the motivation is solid, and the method is novelty.
2. The paper also conducts a thorough comparison with various methods using multiple test datasets.

**Limitations:**

1. Why there is no 3D points of GT in Figure4?
2. Failure cases are missing.
3. Speed evaluations are missing.

**Suitability:**

3

---

### Official Review · Reviewer_oszg · 2024-05-25

**Rating:** 3
**Confidence:** 3

**Summary:**

This study introduces a novel self-supervised category-level object 6D pose estimation method, based on part-level reconstruction and coarse-to-fine correspondence.By analyzing different parts of objects in detail and progressively refining the correspondence, this method not only enhances the accuracy of pose estimation but also improves its adaptability in complex scenarios. This work showcases the effectiveness of combining part-level analysis and fine-grained supervision to advance self-supervised pose estimation techniques. Experiments on the Wild6D and Real275 datasets demonstrate performance improvements over existing self-supervised methods.

**Strengths:**

+ The integration of part-level reconstruction into 6D pose estimation tasks is an intriguing approach  This method provides a detailed understanding of object geometry, enhancing the accuracy of pose estimations.
+ The optimization of Coarse-to-Fine Correspondence through a two-stage feature point matching process effectively alleviates the multi-solution problem in object pose estimation, significantly improving the robustness and precision of the results.

**Limitations:**

+ The paper exhibits a somewhat casual tone in its writing. Specifically, lines 42 and 45 lack necessary citations which are crucial for backing the claims made. Additionally, the use of "regions" in line 118 is ambiguous and lacks clear definition, while line 121 does not convincingly justify the use of an unsupervised approach. Furthermore, the notations in Figure 2 are confusing and detract from the clarity of the presentation.
+ The paper claims to employ part-level reconstruction; however, it appears to only utilize part-level methods without actually performing reconstruction at the level of individual object parts. This discrepancy between claim and method undermines the paper's integrity.
+ The PSR and CFCP modules seem disjointed. The accuracy of the subsequent pose estimation heavily relies on the precision of the reconstruction, yet the integration between these modules is not adequately addressed, which may impact the overall effectiveness of the proposed approach.
+ There is a noticeable absence of experimental validation for the 3D reconstruction aspect of the study. Including empirical evidence to support the efficacy of the reconstruction process would significantly strengthen the paper’s contributions.
+ While the method is described as state-of-the-art, it does not achieve the best results on the Real275 dataset, and the Wild6D dataset is not the primary dataset used by the GPV-Pose’s main works. This observation suggests that claims of state-of-the-art performance should be made with caution, supported by a more rigorous comparison and discussion regarding the context and selection of benchmarks.

**Suitability:**

2

---

### Meta-Review · Area_Chair_PTmG · 2024-07-02

**Recommendation:** Accept (Poster)
**Confidence:** 2

**Metareview:**

This paper introduces a novel self-supervised category-level object 6D pose estimation method based on part-level reconstruction and coarse-to-fine correspondence.

This paper received mixed final ratings: one weak acceptance and two borderline rejections. However, it is pretty tricky that neither of the two negative reviewers gave final ratings. It seems that the reviewers didn't check the rebuttal document. Considering the quality of the paper and the provided rebuttal document, I tend to accept this paper. It is unfair that the reviewers rejected this paper without final ratings and justifications.